# Botulinum Toxin—A High-Dosage Effect on Functional Outcome and Spasticity-Related Pain in Subjects with Stroke

**DOI:** 10.3390/toxins15080509

**Published:** 2023-08-18

**Authors:** Domenico Intiso, Antonello Marco Centra, Michele Gravina, Angelo Chiaramonte, Michelangelo Bartolo, Filomena Di Rienzo

**Affiliations:** 1Unit of Neuro-Rehabilitation Unit and Rehabilitation Medicine, IRCCS ‘Casa Sollievo della Sofferenza’, Viale dei Cappuccini 1, San Giovanni Rotondo, 71013 Foggia, Italy; am.centra@operapadrepio.it (A.M.C.); m.gravina@operapadrepio.it (M.G.); chiaramonte.angelo@gmail.com (A.C.); f.dirienzo@operapadrepio.it (F.D.R.); 2Department of Rehabilitation, Neurorehabilitation Unit, HABILITA Zingonia, Ciserano, 24040 Bergamo, Italy; bartolomichelangelo@gmail.com

**Keywords:** spasticity, stroke, botulinum toxin A, high dose, functional outcome, pain

## Abstract

Stroke patients can develop spasticity and spasticity-related pain (SRP). These disorders are frequent and can contribute to functional limitations and disabling conditions. Many reports have suggested that higher doses than initially recommended of BTX-A can be used effectively and safely, especially in the case of severe spasticity; however, whether the treatment produces any benefit on the functional outcome and SRP is unclear. Studies published between January 1989 and December 2022 were retrieved from MEDLINE/PubMed, Embase, and Cochrane Central Register. Only obabotulinumtoxinA (obaBTX-A), onabotulinumtoxinA, (onaBTX-A), and incobotulinumtoxinA (incoBTX-A) were considered. The term “high dosage” indicates ≥600 U. Nine studies met the inclusion criteria. Globally, 460 subjects were treated with BTX-A high dose, and 301 suffered from stroke. Studies had variable method designs, sample sizes, and aims. Only five (55.5%) reported data about the functional outcome after BTX-A injection. Functional measures were also variable, and the improvement was observed predominantly in the disability assessment scale (DAS). SRP pain was quantified by visual analog scale (VAS) and only three studies reported the BTX-A effect. There is no scientific evidence that this therapeutic strategy unequivocally improves the functionality of the limbs. Although no clear-cut evidence emerges, certain patients with spasticity might obtain goal-oriented improvement from high-dose BTX-A. Likewise, data are insufficient to recommend high BTX dosage in SRP.

## 1. Introduction

Stroke is a leading cause of disability worldwide with poor quality of life. Among the symptoms that produce limitations, spasticity is one of the most frequent and representative, since 18 to 43% of post-stroke patients can develop this disturbance [1]. Spasticity is defined as a motor disorder characterized by a velocity-dependent increase in tonic stretch reflexes (muscle tone) with exaggerated tendon jerks, resulting from hyper-excitability of the stretch reflex as one component of the upper motor neuron syndrome [2]. On the other hand, a key role has been attributed to muscle tissue change. Consequently, a derangement of muscular activation and increased resistance around joints due to the modification of visco-elastic muscle components have been suggested [3]. Spasticity is a relevant clinical problem in rehabilitation medicine as it can lead to the development of persistent abnormal posture, pain, muscular-tendon contractures, and bone deformity. These clinical conditions have been associated with poor functional outcomes; hence, spasticity is one of the major sources of disability in several neurological diseases, particularly stroke.

Several pharmacological and non-pharmacological therapies have been proposed for treating spasticity, and among those, botulinum toxin type A (BTX-A) has become the gold standard in the treatment of the focal type [4].

The typical structure of botulinum toxin consists of a metalloprotease light chain (50 kDa) and a heavy chain (100 kDa) linked by a strict disulfide bond. The heavy chain is responsible for toxin internalization into the neurons and the light chain binds to specific proteins involved in acetylcholine release at the neuronal ending. The toxin acts in the cytosol of nerve endings inhibiting the discharge of the acetylcholine-containing vesicles into the synaptic cleft by cleaving the SNAP-25 protein complex and hence impeding the transmission of nerve impulses at the neuromuscular junction. New cellular mechanisms in blocking acetylcholine release have been disclosed. Pirazzini et al. [5] showed that synaptic vesicles possess an active thioredoxin reductase-thioredoxin system that is functional and responsible for the reduction of the interchain disulfide of some botulinum neurotoxin serotypes (A, C, and E). Furthermore, L chain translocation is assisted by the host chaperone Hsp90 [6]. The aim of the treatment is the reduction of muscular tone and the improvement of joint function [7].

For many years seven types of toxins have been recognized from Clostridium, defined as A through G, but several botulinum toxins have been discovered and identified. In 2017, Zhang et al. [8] found in the chromosome of a Clostridium botulinum the gene of a putative neurotoxin displaying significant divergence, with respect to known BTXs, that was responsible for a case of infant botulism. This toxin was considered a new serotype and designated as BoNT/X. The origin of the novel toxins could be due to genome organization, clustering of toxin genes, and toxin amino acid sequence composition [9]. Currently, more than 40 unique BTXs have been identified, which display variable heterogeneity. Since BTX variants are being identified with increasing frequency, in 2017 an ad hoc committee proposed criteria to designate novel toxins and to manage the nomenclature [9,10]. Understanding the cellular mechanisms of BTXs and the discovery of novel types could open new horizons in the use of BTXs, in particular for therapeutic strategies.

Food and Drug Administration (FDA) and European Regulatory Agencies have licensed onaBTX-A (Botox), obaBTX-A (Dysport), and incoBTX-A (Xeomin) and these types of BTX-A have been approved for the treatment of upper limb spasticity (ULS), while only onaBTX-A (Botox) and obaBTX-A (Dysport) have received the approval for the treatment of lower limb spasticity (LLS).

The optimal dosage of BTX-A is determined by the characteristics of the patients as the severity of spastic hypertonia, number, and dimension of muscles involved, chronicity of the condition, age and body mass, previous response to BTX-A injections and concurrent therapy for spasticity [11]. In the USA, doses up to 400 U for onaBTX-A and incoBTX-A, and 500–1000 U for obaBTX-A have been approved for the treatment of ULS, whereas doses of 300–400 U for onaBTX-A, and up to 1500 U for obaBTX-A have been approved for LLS. On the other hand, in Europe, the botulinum dosage injected for single session should not exceed 600 U for onaBTX-A and 1500 U for aboBTX-A [12]. However, BTX-A doses for muscle and total injection dosage for a single session remain unsolved [11]. Moreover, dosage, dilution muscle-specific dosing as well as the type of BTX-A and interventions that are associated with BTX-A infiltration to boost effects are based on the injector decision [13].

In recent years, many reports have suggested that higher doses than initially recommended of BTX-A can be used effectively and safely, especially in the case of severe ULS and LLS [14,15,16]. In this regard, although evidence is insufficient to recommend a high dose of BTX-A in clinical practice, the benefits of this therapeutic strategy may be clinically acceptable in selected patients [17].

One of the main questions concerns is whether in treating post-stroke spasticity, it is possible to obtain an evident functional improvement other than the reduction of spasticity using both recommended doses and high doses of BTX-A. In this respect, the impact of BTX-A high-dosage injection on the improvement of the functional outcome remains debated and controversial.

Even beyond spasticity, botulinum toxin has proved to be efficacious in reducing the pain associated with several neurological conditions [18,19,20,21] as well as pain related to spasticity [22,23]. Several physiological mechanisms have been proposed including the blockade of the cholinergic transmission in the nociceptive system [22], interaction with TRPV1 receptors, and inhibition of substance P, glutamate, and CRGP synaptic release, which are excitatory neurotransmitters that influence pain generation and transmission [24]. However, the effect of BTX-A high dosage on this type of pain remains unclear.

The present paper aimed to investigate the effect of BTX-A high dosage on the functional outcome and spasticity-related pain (SRP) in post-stroke patients.

## 2. Results

The literature search identified 11,914 citations with 429 eligible. Of these, only nine studies [25,26,27,28,29,30,31,32,33] met the inclusion criteria (Figure 1). Of the enrolled studies, three [30,32,33] had mixed samples including also subjects with spasticity resulting from stroke. Globally, 460 subjects were treated with high doses of BTX-A, and 301 suffered from stroke. Four studies with mixed samples were excluded because spasticity was not due to stroke [14,15,34,35]. Investigations had variable method designs, sample sizes, and aims. Only two studies used a randomized controlled design [25,31]. The aim of the investigations was focused mainly on the effects and safety of higher doses of BTX-A than recommended. IncoBTX-A and onaBTX-A were used in 5 and 3 studies, respectively. One study used both onaBTX-A and incoBTX-A formulations [31]. Doses of incoBTX-A ranged from 400 to 1000 U, and doses of onaBTX-A from 400 to 800 U (Table 1).

### 2.1. Functional Outcome

Variable measures of outcome were used including Barthel Index (BI); Functional Independence Measure (FIM); Disability Assessment Scale (DAS); Goal Attainment Scale (GAS); Functional Ambulation Category score (FAC); Visual Analog Scale for Gait Function (VAS GT).

Functional improvement was reported in 5 (55.5%) studies [25,26,28,29,30]. Of these, four studies enrolled only post-stroke patients, and one included mixed samples [30]. Two open-label prospective studies focused on the same group of subjects who were treated with high doses of BTX-A for 2 years [26,29]. Both studies investigated the effect of incoBTX-A doses of up to 840 U (range from 750 to 840 U) injected in muscles of both UL and LL in the same session to reduce multi-level spasticity. The upper limb was injected with a maximum dosage of 540 U, whereas a dosage of 340 U was administered to the lower limbs (range 250–340 U). The latter study reported the effect of high doses in this population after 2 years of treatment. Both studies demonstrated that repeated high doses of incoBTX-A were safe and efficacious in reducing spasticity without adverse events. However, the improvement of the functional outcome was ascertained only by DAS. In this regard, 48% of patients chose the limb position as the main target, followed by pain (24%), hygiene (16%), and dressing (12%). A significant improvement of the DAS score at 30 and 90 days was observed: 1.7 ± 0.6 (CI 95% 1.4–1.9), 1.8 ± 0.6 (CI 95% 1.6–2.1) (*p* < 0.0000), respectively, compared to baseline: 2.6 ± 0.5 (CI 95% 2.4–2.8). Likewise, a significant improvement was observed in DAS score after 2 years of high doses injections of BTX-A: 2.7 ± 0.4 (CI 95% 2.5–2.9); 1.6 ± 0.5 (CI 95% 1.4–1.8); 1.4 ± 0.5 (CI 95% 1.2–1.6) (*p* = 0.0000), at baseline (t0), 30 days after the first injection set (t1) and 30 days after the last injection set (t2), respectively [29].

Baricich et al. [28] performed a retrospective cohort study that enrolled a small sample of 26 post-stroke subjects who showed a significant improvement to DAS, a four-point scale from 0 = no disability to 3 = severe disability, after a mean dose of 676.9 ± 86.3 U of onaBTX-A. Before the BTX-A injection, the patients chose the primary functional target between the four domains of DAS: dressing, limb position, pain, and hygiene. Mean scores to DAS were 2.3 ± 0.5 (CI 95% 2.1–2.5); 1.5 ± 0.6 (CI 95% 1.2–1.7) (*p* < 0.001); 1.8 ± 0.7 (CI 95% 1.5–2) (*p* < 0.05) at baseline, 30 and 90 days, respectively. However, the study aimed to evaluate the efficacy and safety profile of higher doses of onaBTX-A in patients affected by upper and lower post-stroke spasticity.

In the study by Wissel et al. [30], post-stroke patients received three consecutive injection cycles with incoBTX-A 400 U (cycle 1), 600 U (cycle 2), and 600–800 U (cycle 3), respectively, to evaluate the safety (primary objective) and the efficacy of increasing doses. The global assessment of efficacy was evaluated with the Goal Attainment Scale (GAS) for each cycle at the next injection or the end of the study visit. The percentage of patients’ assessment of “very good” or “good” increased progressively from cycle 1 to cycle 3. All groups of patients obtained an improvement of GAS after BTX-A injection, but interestingly 68.6% (96/140; 95% CI [60.5%, 75.7%]) of those who received a dose of ≥700 U achieved >3 (of 4 possible) treatment goals (GAS score > 0) compared to 25.2% and 50.7% of those who received 400 U and 600 U, respectively. Overall, the mean number of goals achieved by each patient was 1.81 (CI 95% 1.59–2.02) in cycle 1 (N = 155), 2.41 (CI 95% 2.18–2.64) in cycle 2 (N = 152), and 3.03 (CI 95% 2.81–3.24) in cycle 3 (N = 140).

Mancini et al. [25] performed a randomized trial to evaluate the effect of three different doses of BTX-A in the treatment of spastic foot. The gait was ascertained by Visual Analog Scale for Gait Function (VAS GF). Fifteen post-stroke subjects for each group received low, medium, and high doses of onaBTX-A, respectively. The group treated with a high dose received a mean dosage of 540 ± 124.2 U. A reduction of spasticity and improvement of gait was observed in all groups after injection. However, the authors detected that the patients in the high-dose group showed a higher reduction in strength that persisted at follow-up (4 weeks), suggesting that reduction of strength in treated and non-treated muscles could result in the deterioration of the gait function.

In a retrospective study, Ianieri et al. [32] assessed the effect of incoBTX-A subdividing the patients into three groups subject to different dosages of BTX-A. Fifty-eight subjects received 700–1000 U from 775.65 ± 30.45 to 986.65 ± 13.67. The results showed a statistical improvement of FIM in 10 patients treated with 100–400 U when the dosage increased up to 700 U.

### 2.2. Spasticity-Related Pain

Only three studies reported the effect of a high dosage of BTX-A on the SRP [25,26,29]. Of these, two studies concerned the same cohort of stroke patients who were followed for 2 years [26,29]. In the former report, the pain was evaluated by visual analog scale (VAS), and a significant reduction of the mean score was observed after BTX-A injections: 5.2 ± 1.3 (CI 95% 4.6–5.9); 2.2 ± 0.9 (CI 95% 1.9–2.6); 2.5 ± 0.7 (CI 95% 2.2–2.8), at baseline, 30 and 90 days, respectively. In the latter study, pain was evaluated by DAS. The authors reported that, during the follow-up, the rate of patients that chose pain as the primary target of DAS increased compared to the baseline: 39% and 24%, respectively, suggesting that the finding could be explained by the reduction in subjective sensation as pain, rigidity, heaviness of spastic limb that occurs with BTX-A treatment [29].

**Table 1 toxins-15-00509-t001:** Studies investigating high doses of BTX-A in patients with post-stroke spasticity.

Study	Design	Patients/Sex/ Age	BTX-A Doses and Guidance/Follow-Up	Functional Measures and SRP Evaluation	Effect on Spasticity-Related Pain	Effect on Spasticity and Functional Outcome
Mancini et al., 2005 [25]	randomized, double-blind, study	N = 45 pts; N = 15 pts with onaBTX-A high dosage; (M 8, F 7) mean age 63.2 ± 10.1	onaBTX-A (Botox) 540 ± 124.2 U; EMG; 4 months	MRC; MAS; VAS GT; GV;	improvement of pain	prolonged effect of BTX on spasticity, GV, gait function, pain, and presence of clonus
Santamato et al., 2013 [26]	prospective	N = 25 12 F, 8 M; age (range 45–71 years)	incoBTX-A (Xeomin) 840 U (ranged from 750 to 840 U) in both UL and LL; UL muscles received a dosage of a maximum of 540 U; 340 U was administered in LL (range 250–340 U); US; 3 months	AS; DAS; GATR; VAS	improvement of SRP was observed for all patients	improvement of disability and muscle tone. Significant decrease evaluated after 30 and 90 days from the treatment (*p* < 0.05) to functional measures
Invernizzi et al., 2014 [27]	case control	N = 11; 5 M, 6 F; age from 44 to 72 years	incoBTX-A (Xeomin) higher 600 U; 12 U/kg (range 600–800); NR	AS > 2; ECG for HRV (RR interval)	N/A	N/A
Baricich et al., 2015 [28]	cohort; retrospective	N = 26; M 13, F 13; mean age 54.7 ± 11.6	onaBTX-A (Botox) 600 IU; 13 pts > 700 IU; mean dose 676.9 ± 86.3 IU; US; 23 pts were treated at both upper and lower limbs; 3 months	MAS; DAS; GAE	N/A	significant reduction of spasticity (*p* < 0.0001). Improvement in DAS and GAE
Santamato et al., 2017 [29]	cohort; prospective	25 pts; 20 (12 F, 8 M); mean age 60.8 ± 7.8	incoBTX-A (Xeomin); 830 U (ranged from 750 U to 830 U) in both upper and lower limb; US; UL received a dosage of a maximum of 560 U and LL a dosage of a maximum of 460 U (ranged from 260 U to 460 U); 2 years	AS; DAS GATR	the rate of patients that chose pain as the primary target of DAS was increased compared to the baseline: 39% and 24%, respectively	improvements as assessed on clinical scales for spasticity (AS), disability (DAS), and global assessment of treatment response (GATR)
Wissel et al., 2017 [30]	prospective, single-arm, dose-titration study	mixed sample N = 155 pts M 104; F 51; mean age 53.7 ± 13.1 N = 132 with stroke; N = 23 other causes ^	incoBTX-A (Xeomin); 400 to 800 IU; 36–48 weeks	AS; REPAS; GAS;	N/A	dosage up to 800 U was associated with increased treatment efficacy, improved muscle tone, and goal attainment
Baricich et al., 2017 [31]	single-blind randomized controlled crossover study design	10 pts; 7 M, 3 F; age 69 ± 10.5	N = 5 onaBTX-A (Botox) 600 U (670 ± 83.67); N = 5 incoBTX-A (Xeomin) (660 ± 89.44); doses below 12 U/Kg	AS; BI; MI; FAC	N/A	N/A
Ianieri et al., 2018 [32]	cohort, retrospective	mixed sample ° N = 120 N = 58 M 28, F 22; mean age 66 ± 3.2	incoBTX-A (Xeomin) N = 58 received 700–1000 U (from 775.65 ± 30.45 to 986.65 ± 13.67); NR; 2 years	AS; FIM; MyotonPRO	N/A	reduction of spasticity and statistical improvement of FIM in 10 patients treated by 100–400 UI when the dosage increased up to 700 U
Chiu SY et al., 2020 [33]	cohort, retrospective	mixed sample N = 68 pts F 43, M 25; N = 24 with spasticity *, N = 44 with dystonia	onaBTX-A (Botox) > 400 U receiving doses up to 800 U (range 425–800); 12 up 86 months	CGIS	N/A	all patients had a reduction of spasticity after the first treatment and the duration of benefit was 8.8 weeks ± 3.1

Legend: N/A = not applicable; EMG = electromyography; US = ultrasonographic guide; AS = Ashworth Scale; BI = Barthel Index; CGIS = Clinical Global Impression Scale; DAS = Disability Assessment Scale; FAC = Functional Ambulation Category score; FIM = Functional Independence Measure; GAE = Global Assessment of Efficacy; GAS = Goal Attainment Scale; GATR = Global Assessment of Treatment Response; GV = Gait Velocity; HRV = Heart Rate Variability; MAS = Modified Ashworth Scale; MI = Motricity Index; MRC = Medical Research Council scale; REPAS = Resistance to Passive Movement Scale; SRP = Spasticity-Related Pain; VAS = Visual Analog Scale; VAS GT = Visual Analog Scale for Gait Function. ^ brain injury, cerebral palsy, brain tumor [30]; ° spasticity due to stroke, traumatic brain injury, multiple sclerosis, spinal cord injury [32]; * spasticity: common etiologies included stroke, traumatic brain injury, and cerebral palsy [33].

Mancini et al. [25] evaluated the patient’s level of satisfaction with the gait function with the VAS GF and the average intensity of lower limb pain with VAS. Four months after the BTX injections, group I (mean BTX-A total dose: 167 U) showed a return to baseline of all the ratings but VAS Pain, which remained significantly improved. A prolonged effect of BTX-A on spasticity, gait function, and pain was observed in patients who received high dosages.

## 3. Discussion

Spasticity and particular spastic clinical pictures might require higher doses of BTX-A than those recommended; hence the need for more tailored treatment options and flexibility in the doses to inject in each single session. Many reports have suggested that higher doses of BTX-A can be used efficaciously in reducing severe focal spasticity in adult post-stroke patients. Although the aim of the treatment is the reduction of muscular tone, the final objective should be directed to obtain a gain in functionality. The present review did not find evidence that this therapeutic strategy unequivocally improves the functionality of the limbs and the global functional outcome. The studies were variable in method design, sample size, aim, functional measures, and outcome. Only two studies were RCTs [25,31], but the aim did not address functional recovery. All enrolled studies were aimed to investigate the efficacy and safety of the BTX-A high dosage.

In line with the principles of the International Classification of Functioning, Disability and Health (ICF), the objective of treating spasticity is its reduction, but mainly the prevention of disabling conditions that may follow it and the improvement of the functionality and the quality of life of subjects suffering from this motor disorder. Indeed, spasticity is a major cause of disability, globally [36]. It manifests in restricted functional mobility and negatively impacts the quality of life causing several problems: discomfort, pain, or sleep disruption associated with hypertonia and/or spasms; the abnormal posture of a limb or the body; the difficulty of care and increased physical burden on the caregiver; medical complications as the risk of contractures and pressure ulcers [37]. Hence, in the rehabilitation process, the treatment of spasticity by reducing muscular hyperactivity and bodily impairments has a key role to obtain a gain in functionality, and consequently an improvement in quality of life and participation.

The reduction of spasticity is widely demonstrated as a consequence of the injection of the recommended BTX-A doses, and reviews have suggested that some oriented-focused movements unequivocally improve after reduced spasticity as a result of BTX-A injection, in particular in the upper limbs [38,39]. Likewise, high doses of BTX-A have also been demonstrated to be effective and safe [16,40,41] and this therapeutic approach is often used in clinical practice and real life, particularly in treating post-stroke spasticity [42,43]. In this respect, our group performed a systematic review to investigate the efficacy and safety of BTX-A high dosage in treating spasticity following central nervous system damage. High doses of BTX-A were used predominantly in post-stroke patients, but the evidence was insufficient to recommend routine use in clinical practice [17].

In the present paper, we addressed the effect of higher BTX-A dosage on the improvement of functional ability and SRP in subjects suffering from stroke. However, evidence that the improvement of functionality both in the upper and in the lower limb after BTX-A injection is not compelling. Only 301 post-stroke subjects were injected by BTX-A high doses and the studies were not designed to evaluate the recovery according to ICF domains. Functional measures were variable and limited to some oriented movements predominantly in the upper limb. Measures of participation and quality of life were not used. The Barthel Index and FIM were reported in only two studies [31,32]. Of these, one had the aim to investigate the influence of high BTX-A dose on the cardiovascular activity of the autonomic nervous system in chronic stroke survivors with spasticity and did not report data about BI [31]. On the other hand, interestingly, Ianieri et al. showed a statistical improvement of FIM in 10 patients of the group receiving 100–400 U of incoBTX-A when the dosage increased up to 700 UI [32]. Moreover, some oriented-focused movements improved after BTX-A high doses injections [26,28,29,30], but the effects on global functionality and participation were lacking or absent.

Several studies have demonstrated that BTX-A has an effect in reducing neuropathic pain following many neurological disorders [18,19,20,21]. Likewise, BTX-A resulted efficacious in the reduction of SRP [22,23]. In the present review, only three studies reported data about this issue [25,26,29]. Of these, two studies concerned the same small group of post-stroke patients, and SRP was not quantified in the same measure. Therefore, there is no evidence to support the use of BTX-A high dose in treating SRP.

Something should be considered when using BTX-A in high doses. Among these, the main concern when treating spasticity is the weakness that may follow botulinum injection, since the hypertonic muscles may give functional support during the rehabilitation process, such as stabilizing the paretic lower limb in maintaining an upright position. Moreover, the reduction of spasticity does not always improve muscle strength and coexisting paresis. Therefore, therapeutic plans must consider a trade-off between the reduction of hypertonia and conservation of the residual motility [44]. Risk compared to functional benefit should be evaluated. In this regard, Mancini et al. [25] suggested that the reduction of strength in treated and non-treated muscles could result in the deterioration of the gait function.

An important question concerns the cost-effectiveness of the use of BTX-A, in particular, whether the benefit gained justifies the cost of BTX-A, in treating post-stroke spasticity. In this respect, studies have demonstrated that botulinum treatment might be a cost-effective healthcare therapeutic strategy for treating post-stroke patients [45], in particular, aboBTX-A adjunct to rehabilitation interventions can produce a higher number of quality-adjusted life years compared to rehabilitation alone [45,46]. However, studies concerning the cost-utility and the cost-effectiveness of high-dose BTX-A in treating post-stroke spasticity should be planned.

In the present paper, the term “high doses” was defined as dosages ≥ 600 U of licensed BTX-A formulations in Europe and the USA. Among BTX-A types, incoBTX-A was mainly used, though this formulation was authorized only for the treatment of spastic ULS. The reason might be due to the structural characteristic of this BTX-A formulation lacking accessory proteins and probably less at risk to develop neutralizing antibodies, hence safer when using higher doses. However, proper investigations should be planned to clarify this issue. Although the occurrence of an adverse event was beyond the objective of this review, almost all studies have not reported major complications after BTX-A high-dose injections.

To date, many questions remain unclear about the use of high-dose BTX-A in treating adult post-stroke patients with spasticity such as the maximum dosage to inject for a single session, the muscles to treat at single or multiple levels, the cost-effectiveness, and the duration effect [17]. Furthermore, the present review highlights that the functional goals within rehabilitation processes to reduce disability and improve quality of life and participation should be identified and quantified by proper measures according to ICF domains.

## 4. Conclusions

High doses of onaBTX-A and incoBTX-A have been injected in treating the spasticity of adult post-stroke subjects. The present review identified nine studies. None of these intended to investigate the effect on functional recovery and SRP. The studies had variable method designs and there was insufficient evidence that high doses of BTX-A improve the global functionality and SRP. Although no clear-cut evidence emerges, certain patients with spasticity might obtain goal-oriented improvement from high-dose BTX-A.

## 5. Materials and Methods

A search of relevant studies was conducted in MEDLINE/PubMed, the Cochrane Central Register of Controlled Trials, CINAHL, and EMBASE and reports published between January 1989 and December 2022 were included. Search terms included “botulinum toxin type A”, “stroke spasticity”, “botulinum toxin high doses”, “botulinum toxin high dosage”, “upper and lower limb spasticity”, “spasticity-related pain”, “post-stroke pain”, “functional outcome”, “functional recovery”. Related terms were combined using the Boolean “OR” and “AND” operators. Congress abstracts/posters were not considered. Studies were included if: (1) adult subjects who suffered from spasticity following stroke were investigated; (2) the sample size included four or more subjects; (3) mixed samples including stroke patients; (4) doses of BTX-A higher than 600 U were injected, regardless of the aim of the study; (5) high doses of BTX-A licensed by the USA and European authorities were injected alone or combined with adjunctive interventions. Studies with mixed samples were excluded if spasticity was not due to stroke or the causes of spasticity were not reported.

The term “high dosage” was defined as neurotoxin doses injected in a single session higher than 600 U for onaBTX-A and incoBTX-A and higher than 1500 U for aboBTX-A toxins.

## Figures and Tables

**Figure 1 toxins-15-00509-f001:**
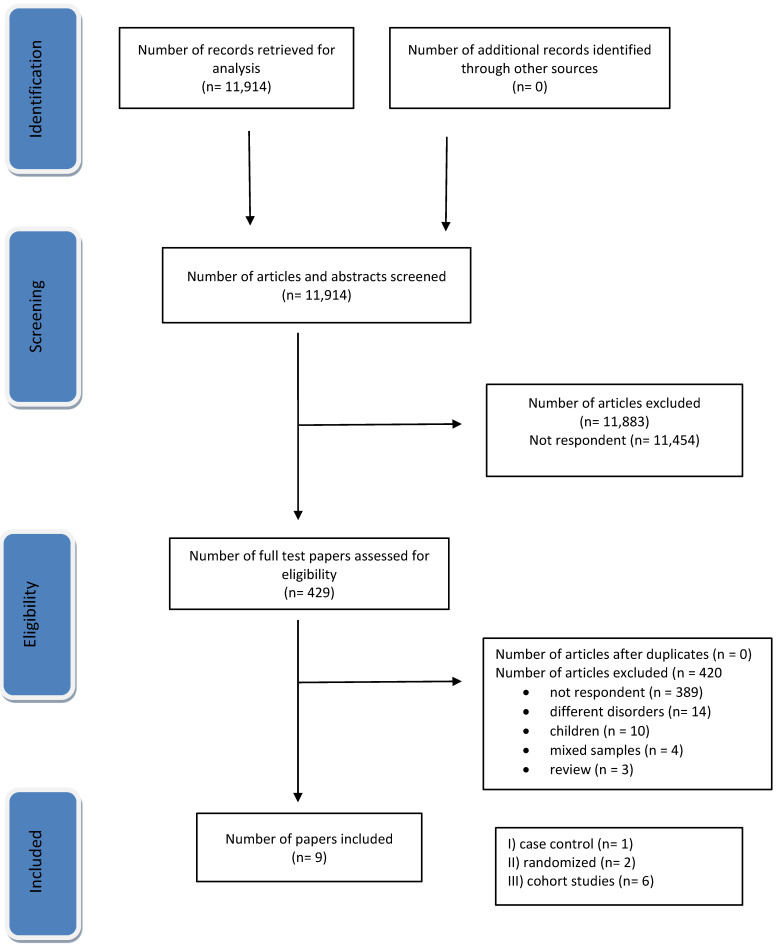
Preferred Reporting Items for Systematic Reviews and Meta-Analyses (PRISMA) diagram depicting the selection of articles for the study.

## Data Availability

Data are available upon request to the corresponding author.

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
