# Peer review of "Botulinum Toxin—A High-Dosage Effect on Functional Outcome and Spasticity-Related Pain in Subjects with Stroke"

_toxins, 2023, doi:10.3390/toxins15080509_

Round 1

Reviewer 1 Report

The paper explores the use of high-dose botulinum toxin type A (BTX-A) in the treatment of post-stroke spasticity, which is a common motor disorder characterized by increased muscle tone and exaggerated tendon jerks. Spasticity can lead to various complications and significantly impact the quality of life of affected individuals. BTX-A is the preferred treatment for focal spasticity, and different formulations have been approved for upper limb and lower limb spasticity. The optimal dosage of BTX-A depends on factors such as the severity of spasticity, muscles involved, chronicity of the condition, and individual patient characteristics. While recommended dosages exist, there is ongoing debate regarding the use of higher doses in certain cases. Some studies suggest that higher doses can be effective and safe, particularly for severe cases of spasticity. However, the impact of high-dose BTX-A on functional outcomes and spasticity-related pain remains uncertain. The paper emphasizes the importance of not only reducing spasticity but also improving functionality and quality of life in the treatment of spasticity. However, the evidence regarding the functional improvement achieved through high-dose BTX-A injections is inconclusive. Functional measures varied across studies, focusing mostly on specific movements rather than overall functionality and participation. The effect of high-dose BTX-A on spasticity-related pain is also not well-established. Considerations when using high-dose BTX-A include the potential weakness that may follow the injection, the trade-off between reducing hypertonia and preserving residual motility, and the cost-effectiveness of treatment. Studies suggest that botulinum treatment can be cost-effective for post-stroke spasticity, but further research is needed to evaluate the cost-utility and cost-effectiveness of high-dose BTX-A specifically. Overall, the paper highlights the need for more research and standardized measures to assess the functional outcomes and efficacy of high-dose BTX-A in post-stroke spasticity treatment, considering the diverse factors involved in individual cases.

Before publication, minor points should be addressed:

-          Authors should describe in more details BoNTs cellular mechanisms of action, e.g. L chain translocation assisted by the host chaperone Hsp90 and disulphide bridge reduction guarantees by the Trx/TrxR system. Please refer to original publications: Pirazzini M. et al., Cell Reports, 2014 and Azarnia Tehran et al., Cellular Microbiology 2019.

-          More recent reviews and articles on serotypes classification and botulinum neurotoxins mechanism of action should be citied and discussed. Please take in consideration that, not only chimera, but more than seven serotypes of BoNTs have been discovered. Please refer to: (i) Zornetta I. et al., Scientific Reports, 2016; (ii) Azarnia Tehran D. et al., Toxins 2018; (iii) Zhang et al., Nature Comm., 2017.

Author Response

I answered to reviewer's comment in uploaded file.

Reviewer 2 Report

The manuscript submitted for evaluation objectively and exhaustively describes the effect of using high doses of botulinum toxins (>600 U) on functional outcome and spasticity-related pain in patients with stroke. The authors rightly noticed on the basis of the meta-analysis of the data that only 2 out of 9 analyzed original papers presented reliable information based on the assumptions of RCTs. There are few literature items that concisely, but exhaustively and objectively, give the opportunity to assess the effectiveness of the use of botulinum toxins in the treatment of spasticity-related pain and functional outcome, even at lower doses. The article seems to be an interesting position for clinicians looking for a reliable compendium of knowledge in the field of the presented topic. Here are just some minor corrections that I feel should be considered in the text:

-I suggest placing references directly after the author's name, e.g. Baricich et al. [number]

-line 244 - "Some considerations should be considered" - you can replace the word "consideration" with something.

Author Response

I thank for the kind consideration and the comments.

I changed the text according to reviewer's suggestions